# Benchmark for Setting ACTH Cell Dosage in Clinical Regenerative Medicine for Post-Operative Hypopituitarism

**DOI:** 10.3390/diseases13040112

**Published:** 2025-04-10

**Authors:** Tatsuma Kondo, Hidetaka Suga, Kazuhito Takeuchi, Yutaro Fuse, Yoshiki Sato, Toshiaki Hirose, Harada Hideyuki, Yuichi Nagata, Ryuta Saito

**Affiliations:** 1Department of Neurosurgery, Graduate School of Medicine, Nagoya University, Nagoya 468-0066, Japan; kondo.tatsuma.b6@s.mail.nagoya-u.ac.jp (T.K.); sato.yoshiki.k3@s.mail.nagoya-u.ac.jp (Y.S.); hirose.toshiaki.w8@s.mail.nagoya-u.ac.jp (T.H.); hide_112@hotmail.co.jp (H.H.); ynagata@med.nagoya-u.ac.jp (Y.N.); saito.ryuta.b1@f.mail.nagoya-u.ac.jp (R.S.); 2Department of Endocrinology and Diabetes, Graduate School of Medicine, Nagoya University, Nagoya 468-0066, Japan; 3Department of Artificial Intelligence Medicine, Graduate School of Medicine, Chiba University, Chiba 260-8722, Japan; yutaro.fuse@gmail.com

**Keywords:** pituitary regenerative medicine, pituitary gland, adrenocorticotropic hormone

## Abstract

Background/Objectives: Our objective is to develop hormone-producing pituitary cells that can function in the same manner as the human body and provide more effective treatments than current hormone replacement therapy. We have already established a technique for generating hypothalamic–pituitary organoids using feeder-free human pluripotent stem cells (hPSCs) and demonstrated their effectiveness in vivo through transplantation into hypopituitary mouse models. To prospectively determine the upper limit of transplanting adenohypophyseal cells into humans, we investigated the human maximum secretion capacity of adrenocorticotropic hormone (ACTH) and growth hormone (GH). Methods: We analyzed data from 28 patients with pituitary adenomas, among whom 16 evinced no abnormality of ACTH secretion and 12 showed no GH secretion on corticotropin-releasing hormone (CRH) and growth hormone-releasing hormone-2 (GHRP-2) stimulation testing. Results: The average ACTH peak value after CRH stimulation tests was 97.2 pg/mL, and the average GH peak value after GHRP-2 stimulation tests was 25.1 ng/mL. Conclusions: These data will likely serve as benchmarks of ACTH and GH secretion when transplanting cultured cells into humans.

## 1. Introduction

The hypothalamic–pituitary system regulates endocrine functions essential for survival. Hypothalamic hormones are secreted into the hypothalamic portal system and directly activate the cell surface receptors in the anterior pituitary. Differentiated pituitary cells secrete six hormones: adrenocorticotropic hormone (ACTH, from corticotroph cells), growth hormone (GH, from somatotroph cells), prolactin (PRL, from lactotroph cells), thyroid-stimulating hormone (TSH, from thyrotroph cells), follicle-stimulating hormone, and luteinizing hormone (FSH and LH, respectively, both from gonadotroph cells). The hypothalamic–pituitary-target axis is crucial for the regulation of pituitary hormone secretion and homeostasis. Corticotropin-releasing hormone (CRH) from the hypothalamus induces ACTH secretion from corticotroph cells in the pituitary gland. ACTH stimulates cortisol secretion from the adrenal glands, forming an example of negative feedback regulation that inhibits the release of CRH and ACTH [1]. Pituitary dysfunction can occur either congenitally or be acquired. The significance of hypopituitarism cannot be overstated, with 45.5 cases per 100,000 population, indicating a substantial burden on public health [2]. Hypopituitarism does not resolve spontaneously, and its treatment primarily relies on hormone replacement therapy (HRT) administered orally. Particularly important is the management of adrenal crisis, a potentially life-threatening complication resulting from inadequate adrenal steroid replacement in adrenocorticotropic hormone (ACTH) deficiency [3]. On the other hand, excessive adrenal steroid replacement can lead over time to iatrogenic Cushing’s syndrome. The current treatment modality of HRT thus fails to adequately track the finely tuned hormonal fluctuations of the body, potentially leading to recurrent slight deficiencies or excesses [4], which may contribute to a greater risk of sudden death than that of healthy individuals.

Growth hormones (GHs) primarily promote somatic growth during childhood and are involved in various aspects of metabolic regulation during adulthood. Children with insufficient GH secretion experience less linear growth and exhibit shorter stature than their peers. Adults with GH deficiency develop alterations in body composition and metabolic abnormalities such as dyslipidemia, fatigue, reduced stamina, and decreased concentration, all of which negatively impact their quality of life. Increased dyslipidemia, thickening of the carotid intima–media, and a higher incidence of non-alcoholic fatty liver disease have been reported, all of which increase the risk of atherosclerosis [5]. GH replacement therapy is administered for these conditions; however, adherence is often suboptimal, particularly among adults, due to the necessity for self-administration via injection [6]. Our final objective is to develop pituitary hormone-producing cells that respond to environmental cues, such as those that the human body supplies, and to offer therapy that is more effective than the current HRT.

Human pluripotent stem cells (hPSCs) are broadly categorized into two distinct types based on their origin: human embryonic stem cells (hESCs) [7] and induced pluripotent stem cells (iPSCs) [8]. These cells possess the remarkable capacity to differentiate into derivatives of all three germ layers. Extensive research has elucidated numerous methodologies to direct their differentiation into various specialized cell types, including those of the central nervous system [9,10], cardiopulmonary system [11,12], gastrointestinal tract [13,14], and musculoskeletal system [15,16].

Our previous studies have focused on developing a differentiation protocol for producing pituitary–hypothalamic organoids (PHOs) utilizing human pluripotent stem cells (hPSCs). During embryonic development, the pituitary gland originates from the oral ectoderm in contact with the adjacent hypothalamus. Throughout this process, interaction between the oral ectoderm and hypothalamus is a critical feature of pituitary cell development, although the precise mechanism remains incompletely elucidated. This differentiation process is tightly regulated temporally and spatially by various growth factors such as Bone Morphogenetic Protein-4 (BMP4), sonic hedgehog, fibroblast growth factor, and wingless/integrated signaling [17,18], followed by transcription factors. Various methodologies have been proposed to induce pituitary cells by mimicking embryonic pituitary differentiation. The efficient differentiation of cortical progenitors from mouse embryonic stem cells (mESCs) was initially achieved through a serum-free suspension culture, which is called a serum-free floating culture of embryoid body-like aggregates (SFEB) [19]. This culture method dissociates mESCs to form spontaneous floating aggregates, which subsequently differentiate into cortical progenitor cells that express brain factor 1 and neural progenitor cells. Furthermore, a modified serum-free floating culture of embryoid-like aggregates with quick reaggregation (SFEBq) culture method [20] utilizing a strictly chemically defined medium without growth factors enables differentiation into hypothalamic progenitors [21]. After approximately 25 days of SFEBq culture, mESCs differentiate into mature hypothalamic neurons through hypothalamic progenitors expressing the retinal and anterior neural fold homeobox (Rax). Based on these findings, we attempted to induce pituitary differentiation by replicating the interaction between the hypothalamus and oral ectoderm in vitro [22]. Increasing the number of cells in the organoids enhanced endogenous BMP4 signaling, which induced oral ectodermal cells in the outer layer of the hypothalamic organoids. This simultaneous induction of the oral ectoderm and hypothalamus reproduces the interaction between these tissues and leads to the self-organization of pituitary hormone-producing cells. In 2016, we advanced mouse ES cells into human ES cells [23]. In 2020, we successfully generated pituitary–hypothalamic tissue from human iPS cells with a success rate of approximately 30% [24]. Three key points emerge from these methods: (1) similar to the development of a fetus in three dimensions, pluripotent stem cells such as ES cells or iPS cells must be cultured in three dimensions from an early undifferentiated state; (2) recapitulating fetal development in vitro is essential, with the hypothalamus induced simultaneously with the pituitary; and (3) the interaction between these tissues must induce well-differentiated tissues. We have successfully differentiated functional pituitary ACTH cells. Several researchers have developed anterior pituitary cells from hPSCs utilizing two-dimensional (2D) adherent culture methods [25,26]. Although functional pituitary hormone-producing cells can be induced by both methods, there are notable differences. Firstly, the induction period was shorter in the two-dimensional culture. It required 30 days to induce pituitary hormone-producing cells in a two-dimensional culture and 70 days in the three-dimensional culture. Secondly, hypothalamic cells were simultaneously induced in three-dimensional organoids. This simultaneous induction of the hypothalamus and pituitary gland replicates the functional regulation of the hypothalamic–pituitary axis [24]. Thirdly, there are differences in the differentiation tendencies of the pituitary hormone-producing cells. ACTH secretion is higher in three-dimensional culture, whereas GH secretion is higher in two-dimensional culture [23,24,25,26].

We have been working continuously to refine differentiation-induced pituitary cells in three-dimensional culture into a technique suitable for clinical use in humans. In 2023, we established a feeder-free pituitary–hypothalamic organoid (PHO) generation technique using hPSCs with clinical application in mind, achieving nearly 100% success in producing pituitary–hypothalamic tissues, including ACTH-producing cells [27]. PHOs transplanted into a mouse model of hypopituitarism successfully engrafted, improved blood ACTH levels, and responded to stimulatory and inhibitory signals [28]. Comparable therapeutic outcomes were observed when purified pituitary cell populations, isolated from PHOs, were employed for transplantation. To develop a novel cell transplantation therapy, it is essential to conduct clinical trials safely, which requires non-clinical safety evaluations using products with assured quality. We have previously developed a differentiation protocol that adheres to standards for raw materials of biological origin and have demonstrated the successful purification and selective enrichment of functional pituitary cells [27]. In addition, we have conducted non-clinical efficacy studies using these cell products in mouse models. Our future research agenda includes the accumulation of comprehensive non-clinical data to support the design of human clinical trials. This will involve studies utilizing large animal models to optimize transplantation techniques, assess immunosuppressive and adjunctive therapies, and validate safety and efficacy parameters. In parallel, we aim to establish a scalable manufacturing platform utilizing clinical-grade iPSCs, thereby laying the foundation for the eventual clinical realization of pituitary cell-based transplantation therapies.

Although hPSC-derived pituitary organoids hold promise for regenerative medicine in patients with hypopituitarism, several challenges to their implementation remain, one of which is determining the appropriate cell dosage for transplantation into humans. Because transplanted cells can adjust their secretion in response to the environment, the baseline secretion levels may not reflect the capacity of the cells. It is desirable to perform stimulation tests to evaluate the secretory capacity of transplanted cells. We conducted CRH stimulation testing in mouse experimental models to assess the maximum secretory capacity of ACTH; however, the maximum secretory capacity of ACTH in humans remains unknown. In this study, we retrospectively investigated the maximum secretory capacity of ACTH and GH in patients with pituitary adenomas who exhibited normal ACTH and GH secretion in response to CRH and growth hormone-releasing peptide-2 (GHRP-2) stimulation tests.

## 2. Materials and Methods

Among 40 patients who underwent surgery at the Department of Neurosurgery, Nagoya University School of Medicine, between 2020 and 2022 for pituitary adenomas, 16 showed normal ACTH responses in CRH stimulation tests conducted before and after the initial surgery, and 12 showed normal GH responses in GHRP-2 stimulation tests. We assessed ACTH peak value in CRH stimulation tests and GH peak value in GHRP-2 stimulation tests. The CRH stimulation test was conducted in the morning (no later than 10:00 A.M.), in a fasting state. After approximately 30 min of rest in the supine position, the test was administered while maintaining the same posture. A total of 100 μg human CRH (Human CRH for Intravenous Injection “Tanabe”, Nipro ES Pharma Co., Ltd., Osaka, Japan) was reconstituted in 1 mL of diluent and administered intravenously over approximately 30 s. Plasma ACTH levels were measured prior to stimulation and at 30, 60, 90, and 120 min post-stimulation. Serum cortisol levels were measured prior to stimulation and at 30, 60, 90, and 120 min post-stimulation. Similarly, the GHRP stimulation test was conducted in the morning (no later than 10:00 A.M.), in a fasting state. Following approximately 30 min of rest in a supine position, the test was administered while maintaining the same posture. A total of 100 μg of GHRP-2 (GHRP Injection Kaken 100^®^, Kaken Pharmaceutical Co., Ltd., Tokyo, Japan) was reconstituted in 10 mL of physiological saline provided with the preparation and administered intravenously over approximately 30 s. Serum GH levels were measured prior to stimulation and at 15, 30, 45, and 60 min post-stimulation. Plasma ACTH/serum cortisol levels and serum GH/blood IGF-1 levels obtained from the aforementioned methods were determined using electrochemiluminescence immunoassay (ECLIA) kits clinically utilized in Japan (SRL Inc., Tokyo, Japan). The criteria for a normal response in the stimulation tests were defined according to the guidelines of the Japan Endocrine Society. Specifically, we considered ACTH levels rising to 1.5 times or more of the pre-administration level and reaching 30 pg/mL or higher after CRH stimulation, with cortisol levels rising to 1.5 times or more of the pre-administration level and reaching 15 μg/dl or higher, as indicative of normal adrenal function. Additionally, we considered normal GH secretory function serum GH peak values, measured every 15 min over 60 min after GHRP-2 administration, to be 9 ng/mL or higher, with insulin-like growth factor 1 (IGF-1) levels within the normal range. In the human CRH stimulation test, plasma ACTH levels attained their maximum at 30 or 60 min post-stimulation in all subjects. For statistical analysis, the peak plasma ACTH value following stimulation was utilized. In the current GHRP-2 stimulation test, serum GH levels reached their maximum 15 or 30 min post-stimulation. The peak plasma GH value following stimulation was employed for statistical analysis.

## 3. Results

We investigated the demographics of 16 patients with normal ACTH responses in CRH stimulation tests and 12 patients with normal GH responses in GHRP-2 stimulation tests (Appendix A). Among the 16 with normal ACTH responses, the median age was 63 years (IQR 55.8–69), with a baseline ACTH level of 24.2 pg/mL (18.0–30.7) and a baseline cortisol level of 10.6 μg/dL (8.0–11.7) (Table 1). In the 12 with normal GH responses, the median age was 58.5 years (IQR 51.3–65.8), with a baseline GH level of 0.58 ng/mL (0.18–1.25) and IGF-1 levels of 140.5 μg/dL (118.9–149.8) (Table 2). Figure 1 illustrates the ACTH peak value in CRH stimulation tests and the GH peak value in GHRP-2 stimulation tests. The median ACTH peak value was 95.65 pg/mL (range 82.6–113.5, n = 16), and the median GH peak value was 17.70 ng/mL (range 13.5–39, n = 12). The mean ACTH peak value on CRH stimulation testing was 97.2 pg/mL, and the mean GH peak value on GHRP-2 stimulation testing was 25.1 ng/mL.

## 4. Discussion

In Japan, it is not standard practice to conduct hormone stimulation tests on healthy adults. The current study was a retrospective analysis, and hormone stimulation tests were not performed on healthy individuals. In this study, we investigated the maximum secretion capacity for ACTH and GH in 16 patients who, before and after initial surgery, demonstrated normal ACTH responses in CRH stimulation tests and 12 patients who, before and after initial surgery, demonstrated normal GH responses in GHRP-2 stimulation tests. This study proposes that the identified maximum secretion capacity will serve as a crucial benchmark for determining the upper limit of pituitary cell dosage in future regenerative therapy through pituitary cell transplantation.

Studies of ACTH and GH secretion capacity exist. Plasma ACTH levels in non-anesthetized, unstimulated rats reportedly increase immediately after the onset of stress, attaining nearly maximum levels (932 pg/mL) 5 min thereafter, and this elevated level persists for 60 min before gradually decreasing [29]. Although plasma GH levels vary by sex and age, female rats generally exhibit higher plasma GH concentrations than males, with mean concentrations of 94.2 ± 17.3 ng/mL in females and 54.9 ± 12.0 ng/mL in males [30]. In aged rats, the peak GH concentration reportedly reaches 200–300 ng/mL [31]. In adult men, the mean peak ACTH value during a CRH stimulation test (nine subjects) was 56.8 ± 3.2 pg/mL [32], and the mean peak GH value during a GHRP-2 stimulation test (eight subjects) was 68.7 ± 15.5 ng/mL [33]. In our study, the mean peak ACTH value on CRH stimulation testing was 97.2 pg/mL, and the mean peak GH value on GHRP-2 stimulation testing was 25.1 ng/mL. The average ratio of the peak values of ACTH to GH after stimulation was 1:258.2. These values may serve as a benchmark for ACTH and GH secretion levels in future human cell transplantation.

As a limitation, while stratification by age for GH would ideally be necessary, this study did not stratify its subjects’ data due to the limited number of cases and instead treated them as a single group for statistical analysis. Additionally, data from cases with normal ACTH secretion and decreased GH secretion were included in the ACTH analysis, while cases with decreased ACTH secretion and normal GH secretion were included in the GH analysis. Furthermore, the patient population in this study excludes individuals with ACTHomas or GHomas. Consequently, while the peak ACTH and GH values presented in this study may decrease following surgery, it is unlikely that they would increase. In essence, the values shown may be comparatively low; however, this does not pose a significant issue when considering them as the maximum dosage thresholds for future pituitary regenerative medicine. This approach aims to prevent excessive dosing. Further accumulation of cases is warranted in future studies.

## 5. Conclusions

We have provided indices of human maximum secretion capacity for ACTH and GH. This will likely serve as a benchmark for transplanting human pituitary hormone-producing cells into humans.

## Figures and Tables

**Figure 1 diseases-13-00112-f001:**
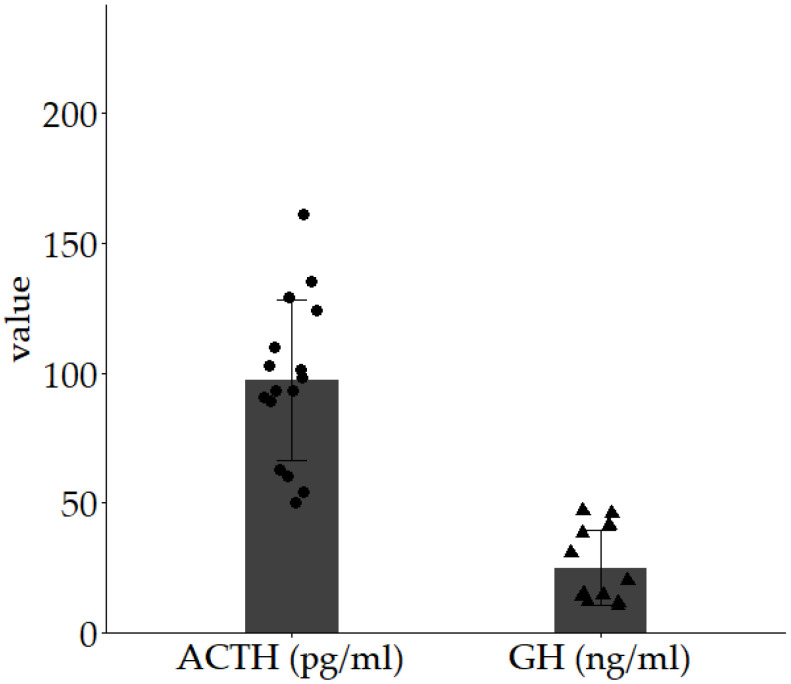
Peak values of ACTH on CRH stimulation testing and peak values of GH on GHRP-2 stimulation testing.

**Table 1 diseases-13-00112-t001:** Background of patients with normal ACTH response.

	Median (IQR)
Age	63 (55.8–69)
Sex (M:F)	11:5
ACTH (pg/mL)	24.2 (18.0–30.7)
Cortisol (µg/mL)	10.6 (8.0–11.7)
Tumor type	NFPA 14, TSHoma 2

**Table 2 diseases-13-00112-t002:** Background of patients with normal GH response.

	Median (IQR)
Age	58.5 (51.3–65.8)
Sex (M:F)	5:7
GH (ng/mL)	0.58 (0.18–1.25)
IGF-1 (ng/mL)	140.5 (118.8–149.8)
Tumor type	NFPA ^1^ 9, TSHoma ^2^ 2, PRLoma ^3^ 1

^1^ Non-functional pituitary adenoma, NFPA. ^2^ Thyroid-stimulating hormone-secreting adenoma, TSHoma. ^3^ Prolactinoma, PRLoma.

## Data Availability

The raw data supporting the conclusions of this article will be made available by the authors upon request.

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
