# Peer review of "Benchmark for Setting ACTH Cell Dosage in Clinical Regenerative Medicine for Post-Operative Hypopituitarism"

_diseases, 2025, doi:10.3390/diseases13040112_

Round 1

Reviewer 1 Report (Previous Reviewer 2)

Comments and Suggestions for Authors

In this study, authors determinates GH and ACTH values but the size sample is low again. These are tumors samples from Neuroradiology Dept. Although this time, the information about regenerative medicine protocols in vitro has been removed, it also appaers in the title but the manuscript does non inform about regenerative medicine findings by these authors.

The values for hormons among men and womens with tumors are divided but it is not enough to conclude anything with this low sample size. In addition, hormons were not determinated before/after treatments; moreover, the posiblle future infusion of iPS in patients would requiere the ethitical approval from a clinic hospital. So, these results are not enough and statistical analysis is not clear again. This reviewer can not see the p values for hormons althogh they divided patients among men and women.

I don`t share the conclussion of this study and the size sample should be  again for a solid conclusion of these preliminary findings. The determination of hormons after a possible iPS treatment should require the previous approval by a clinic ethical commite.

This manuscript contains metodological flats that does not allow its aceptation. The size simple is low and the clinical relevance of these findings, at least, is not clear without data with iPS treatments in these patients. Although this time , plasma GH or ACTH can be influeenced by sex and age, the low size sample does not allow to conclude their clinical relevance. In addiiton, the possible effect of iPS treatment able to normalize them to control values can be tested here. They does not evalaute their hormone levels after iPS treatment in patients, which require a previous ethical commite approval before taking samples for further evaluation of iPS against tumors.

Shall you describe the statistical analisis done in this study?

Comment to authors

I also reply to their author`s comments.

Thanks¡

Response 1. The present study did not include data on patients who underwent cell transplantation. Pituitary cell transplantation therapy has not yet been achieved, and is currently under development for future implementation.

Yes, but why you include regenerative medicine in the title again?

This study provides statistical data to derive reference figures for transplantation. The progress of our research to date is described below: We have established a technique to generate pituitary-hypothalamic organoids (PHOs) using feeder cell-free human pluripotent stem cells (hPSCs) for use in human clinical practice and have demonstrated the efficacy of cell transplantation therapy of pituitary organoids in a mouse model of hypopituitary function.

My response. Correct but this evidence must be confirmed at the clinical level and this is not the case in your study.

Currently, we are conducting cell transplantation experiments in large mammals to verify the efficacy of cell transplantation therapy. To prepare for future pituitary cell transplantation therapy in humans, we aimed to determine the optimal dosage of cells for transplantation.

My response. The levels of these hormons are not necesarly the same in rodent models and humans

 As the transplantable cells we created possess the ability to adjust their secretion in response to the surrounding environment, the amount of basic ACTH secreted by the transplantable cells is not necessarily proportional to their maximum secretory capacity.

Correct, this is a problema in case further transplantation in humans because hormon levels are not comparable between mices and humans.

For instance, even if we transplant twice the number of cells, basal secretion is maintained within a certain range owing to feedback mechanisms. To accurately measure the ability of transplanted cells, it is advisable to include a stimulation test to assess the maximum secretory capacity, thereby avoiding potential overdosing of the cell preparations. The maximum secretory capacity of ACTH in the transplanted cells was evaluated by performing the CRH stimulation test in an experimental mouse model. However, the maximum secretory capacity of ACTH in humans remains unknown.

My response. In fact, the last evidence, does not allow the clinical use of these iPS for transplants without taking previous preliminar data in humans

 In the present study, we conducted a retrospective analysis of the maximal secretory capacity of ACTH and GH in patients with pituitary adenomas who were determined to have no impairment in ACTH and GH secretory capacity after undergoing CRH and GHRP-2 loading tests. The results indicated that the mean ACTH peak value in the CRH stimulation test was 97.2 pg/ml and the mean GH peak value in the GHRP-2 stimulation test was 25.1 ng/ml. The mean ratio of the ACTH to GH peak values after stimulation was 1:258.2. We propose that these data will serve as realistic indicators for future dose testing in human clinical practice.

My response. These data can be different depending of age and sex in patients.l I don`t agree with you

My Decision is reject the manuscript in the present form without possibility of resubmision

Thanks

Author Response

Dear Reviewer 1,

Thank you very much for your kind feedback. I sincerely appreciate your efforts in reviewing and helping improve the quality of our manuscript through your insightful edits.

However, there appears to be a misunderstanding concerning the purpose and content of our study. The data presented in the manuscript do not pertain to patients who have received transplantation of pituitary hormone-producing cells derived from iPS cells. As of now, pituitary cell transplantation therapy has not yet reached the stage of clinical application and remains under development.

This study is a retrospective investigation of the maximum ACTH and GH secretory capacity in normal humans. The intention is to provide standard reference values that may serve as a benchmark for determining appropriate dosage when pituitary cell transplantation is implemented in future human clinical applications.

As acknowledged in the manuscript, the limited number of cases is a recognized limitation of the study. we are continuing our efforts to accumulate further clinical cases for future research.

We have previously addressed similar comments during Revision 1 Round 1. Therefore, we regret that we may not have fully understood the intent of your current remarks, despite our previous response.

We appreciate your understanding in this matter and thank you once again for your constructive support.

Best wishes,

Hidetaka Suga, M.D., Ph.D.

Reviewer 2 Report (Previous Reviewer 1)

Comments and Suggestions for Authors

Thank you for the detailed explanations, and the edits to the manuscript. 

Author Response

Dear Reviewer 2,

Thank you very much for your kind feedback. I sincerely appreciate your efforts in reviewing and helping improve the quality of our manuscript through your insightful edits.

Your support is truly valuable, and I am grateful for your guidance throughout this process.

Best wishes,

Hidetaka Suga, M.D., Ph.D.

Reviewer 3 Report (New Reviewer)

Comments and Suggestions for Authors

This is a well-written and relevant manuscript that addresses an important aspect of regenerative medicine for hypopituitarism. The introduction provides strong background and justification for the study, and the results are clearly presented and supported by appropriate methodology. The findings offer meaningful benchmarks for future clinical application in ACTH and GH cell transplantation. Overall, the manuscript is of high quality. No major revisions are required

Author Response

Dear Reviewer 3,

Thank you very much for your thoughtful and encouraging comments regarding our manuscript. We sincerely appreciate your recognition of the relevance and clarity of our study, as well as your kind words about the quality of our work. Your positive feedback on the introduction, methodology, and the significance of our findings for future clinical applications in ACTH and GH cell transplantation is truly encouraging for our team.

We are grateful for your time and efforts in reviewing our manuscript, and we will continue striving to contribute meaningful research to the field of regenerative medicine.

Best wishes,

Hidetaka Suga, M.D., Ph.D.

This manuscript is a resubmission of an earlier submission. The following is a list of the peer review reports and author responses from that submission.

Round 1

Reviewer 1 Report

Comments and Suggestions for Authors

The authors have submitted a manuscript wherein they analyze and report the maximum secretory capacity of apparently normally functioning pituitary glands. The purpose of this study is to define the secretory characteristics of the normal human pituitary gland. With these data, the authors propose provide benchmarks for future studies where they plan to implant human pluripotent stem cell derived pituitary organoids as tissue replacement therapy. The authors investigated the ACTH and GH peak values in patients that underwent surgery for pituitary adenomas. The authors then identified the maximum secretory capacity for ACTH and GH by analyzing the peak levels following stimulation with CRH and GHRP-2 stimulation respectively. 

Conceptually, the study is of great importance, however, the implementation in the manuscript will need clarification.

1.     Why did the authors choose only patients that underwent surgeries for pituitary adenomas? Would these patients not have subtle hormone secretory problems that make the determination of maximum capacity fraught with errors? Please explain the rationale in detail. 

2.     Timing: The authors need to analyze the report the timing of the peak levels. Without the timing datapoints, interpretation of the results is challenging. 

3.     Was there an overlap in the 12 and 16 patients whose data are reported?

4.     What about the rest of the patients (12?)? Why were their data not included in the analysis?

5.     What is the minimum secretory capacity that the authors still consider ‘normal’?

6.     Most importantly, did the authors review the extensive literature on CRH and DDAVP stimulation data? The best results to calculate the ‘normal’ maximum secretory capacity may be to analyze the peaks in non-tumor cases. 

7.     Can the authors give us a sense of the absolute values (Molar or in nanograms) of the secretion that they expect the normal glands to secrete? Would they consider providing models for secretion rates based on the expected dilutions (approximately 5 L of blood) in their patients? I would like to learn that the normal pituitary gland will secrete a total of x ng ACTH following a stimulation with y micrograms of CRH. 

Author Response

Reviewer 1

Comments 1. Why did the authors choose only patients that underwent surgeries for pituitary adenomas? Would these patients not have subtle hormone secretory problems that make the determination of maximum capacity fraught with errors? Please explain the rationale in detail. 

Response 1. Numerous patients targeted by pituitary regenerative medicine are hypothesized to have developed pituitary dysfunction after surgical intervention for hypothalamic-pituitary tumors. The aforementioned comments elicited the following responses. In Japan, it is not standard practice to conduct hormone stimulation tests in healthy adults. The current study was a retrospective analysis, and hormone stimulation tests were not performed in healthy individuals. The patient population in this study excluded individuals with ACTHomas and GHomas. Consequently, while the peak ACTH and GH values presented in this study may decrease following surgery, it is unlikely that they would increase. In essence, the values shown may be comparatively low; however, this does not pose a significant issue when considering them as maximum dosage thresholds for future pituitary regenerative medicine. The aim of this approach is to prevent excessive dosing. The aforementioned two points were incorporated into the Discussion and Limitations sections. The criteria for a normal response in the stimulation tests were defined according to the guidelines of the Japan Endocrine Society. Specifically, we considered as indicative of normal adrenal function ACTH levels rising to 1.5 times or more of the pre-administration level and reaching 30 pg/ml or higher after CRH stimulation, with cortisol levels rising to 1.5 times or more of the pre-administration level and reaching 15 μg/dl or higher. Additionally, we considered normal GH secretory function serum GH peak values measured every 15 minutes over 60 minutes after GHRP-2 administration to be 9 ng/ml or higher, with insulin-like growth factor 1 (IGF-1) levels within the normal range. These details are also elucidated in the Methods section. 

Comments 2. Timing: The authors need to analyze the report the timing of the peak levels. Without the timing datapoints, interpretation of the results is challenging. 

Response 2. In the human CRH stimulation test, plasma ACTH levels attained their maximum at 30 or 60 min post-stimulation in all subjects. For statistical analysis, the peak plasma ACTH value following stimulation was utilized. In the current GHRP-2 stimulation test, serum GH levels reached their maximum 15 or 30 min post-stimulation. The peak plasma GH value following stimulation was employed for statistical analysis. These points were added to the Methods section.  

Comments 3. Was there an overlap in the 12 and 16 patients whose data are reported?

Response 3. Furthermore, eight patients were included in both groups. These eight patients did not present with ACTHomas or GHomas and demonstrated normal responses in each stimulation test. Consequently, they were incorporated into the statistical analyses for each test. This point has been documented in the Limitations section. 

Comments 4. What about the rest of the patients (12?)? Why were their data not included in the analysis?

Response 4. The criteria for normal response in the stimulation tests were defined according to the guidelines of the Japan Endocrine Society. Specifically, we considered as indicative of normal adrenal function ACTH levels rising to 1.5 times or more of the pre-administration level and reaching 30 pg/ml or higher after CRH stimulation, with cortisol levels rising to 1.5 times or more of the pre-administration level and reaching 15 μg/dl or higher. Additionally, we considered as indicative of normal GH secretory function serum GH peak values measured every 15 minutes over 60 minutes after GHRP-2 administration being 9 ng/ml or higher, with insulin like growth factor 1 (IGF-1) levels within normal range. In instances where either the ACTH response or GH response did not meet the aforementioned criteria, only cases with a normal response were included in the analysis. This limitation is addressed in the Limitations section.

Comments 5. What is the minimum secretory capacity that the authors still consider ‘normal’?

Response 5. The criteria for normal response in the stimulation tests were defined according to the guidelines of the Japan Endocrine Society. The definition of a normal response is delineated in the Methods section. Accordingly, the plasma ACTH level is 30 pg/mL or higher at the CRH loading apex. The serum GH level is ≥ 9 ng/mL at the apex of the GHRP2 loading. 

Comments 6. Most importantly, did the authors review the extensive literature on CRH and DDAVP stimulation data? The best results to calculate the ‘normal’ maximum secretory capacity may be to analyze the peaks in non-tumor cases. 

Response 6. As previously noted, data on non-tumor cases that exhibit a normal response to the stimulation test are not available. It is stated in the Methods section that the study was conducted on oncological surgery cases. In the context of pituitary regenerative medicine, it is presumed that the majority of target patients will experience impaired function following pituitary tumor surgery, which aligns with the objectives of the present study. To address this point, the title has been revised accordingly. Furthermore, as elucidated in the discussion, the data pertaining to CRH stimulation tests are limited to studies conducted on rats and findings from the 1980s. While acknowledging the limitations of the present data, it is noteworthy that these findings represent recent human data, which enhance their value in the field. 

Comments 7. Can the authors give us a sense of the absolute values (Molar or in nanograms) of the secretion that they expect the normal glands to secrete? Would they consider providing models for secretion rates based on the expected dilutions (approximately 5 L of blood) in their patients? I would like to learn that the normal pituitary gland will secrete a total of x ng ACTH following a stimulation with y micrograms of CRH.

Response 7. Analysis of our data suggests that the normal pituitary gland would secrete a total of 486 ng of ACTH following stimulation with 100 μg/ml CRH; however, the accuracy of this estimation remains uncertain due to the approximate nature of fluid volume measurement. Although it may be feasible to measure ACTH in vitro using an animal model by extracting the pituitary gland, such an approach is not viable in human subjects. Future research endeavors will aim to address these questions through the accumulation of data on plasma ACTH levels in healthy subjects and more precise measurements of the fluid volume in which ACTH is distributed.

Reviewer 2 Report

Comments and Suggestions for Authors

Comments to authors

In this study, authors include this coments but values for GH, IGF-1 and ACTH fom a low number of patients with tumors from Neuroradiology Dept without any connextion with the fied or evidences in the regenerative medicine. These findings and thecnics are interesting but are unfocus with the mean of their study. In fact, they  only showed values for hormons in patients with tumore without stem cell therapy. So, these results are not enough good explained for me and are not focus in regenerative medicine.

In fact, these hormons values could be totally different in stem cell therapy-treated patients with these tumors. However, these data are not showed in the present manuscript.

However, this review does not seem this conexion. The content of this manuscript is stange because they show a protocol in vitro for regenerative medicine but findings are in human samples with GH and ACTH without any logical conexión.

I don,t share the conclussion of this study and the size sample should be extended before concluding their conclusssion.

They comment that PHOs transplanted into a mouse model of hypopituitarism successfully engrafted, improved blood ACTH levels, and responded to stimulatory and inhibitory signals. However, this study limits to measure GH, ACTH and IGF-1 in a low number of patients with tumors without any treatment with stem cell therapy.

This manuscript contains metodological flats that does not allow its Aceptation. The size simple is low (n=40). In fact, they study the maximum secretion capacity for ACTH and GH in 176 16 patients who before and after initial surgery demonstrated normal ACTH responses in 177 CRH stimulation tests and 12 patients who before and after initial surgery normal GH 178 responses in GHRP-2 stimulation test. Additionally, the tumor type is different (1 and 2). I understand the dificulty to find these simples but there are many flats in this study, including the absence of controls.

The discussion does not analyze the involvement of these findiings in humans with these tumors. Moreover, they indicate plasma GH levels vary by sex and age. In fact, female rats generally exhibit higher plasma GH concentrations than males, with  mean concentrations of 94.2 ± 17.3 ng/ml

However, men and women were measured together with a low size sample.

My Decision is rejected

Author Response

Reviewer 2

Comments 1.

In this study, authors include this coments but values for GH, IGF-1 and ACTH fom a low number of patients with tumors from Neuroradiology Dept without any connextion with the fied or evidences in the regenerative medicine. These findings and thecnics are interesting but are unfocus with the mean of their study. In fact, they only showed values for hormons in patients with tumore without stem cell therapy. So, these results are not enough good explained for me and are not focus in regenerative medicine.

In fact, these hormons values could be totally different in stem cell therapy-treated patients with these tumors. However, these data are not showed in the present manuscript.

However, this review does not seem this conexion. The content of this manuscript is stange because they show a protocol in vitro for regenerative medicine but findings are in human samples with GH and ACTH without any logical conexión.

I don,t share the conclussion of this study and the size sample should be extended before concluding their conclusssion.

They comment that PHOs transplanted into a mouse model of hypopituitarism successfully engrafted, improved blood ACTH levels, and responded to stimulatory and inhibitory signals. However, this study limits to measure GH, ACTH and IGF-1 in a low number of patients with tumors without any treatment with stem cell therapy.

This manuscript contains metodological flats that does not allow its Aceptation. The size simple is low (n=40). In fact, they study the maximum secretion capacity for ACTH and GH in 176 16 patients who before and after initial surgery demonstrated normal ACTH responses in 177 CRH stimulation tests and 12 patients who before and after initial surgery normal GH 178 responses in GHRP-2 stimulation test. Additionally, the tumor type is different (1 and 2). I understand the dificulty to find these simples but there are many flats in this study, including the absence of controls.

The discussion does not analyze the involvement of these findiings in humans with these tumors. Moreover, they indicate plasma GH levels vary by sex and age. In fact, female rats generally exhibit higher plasma GH concentrations than males, with mean concentrations of 94.2 ± 17.3 ng/ml

However, men and women were measured together with a low size sample.

My Decision is rejected

Response 1. The present study did not include data on patients who underwent cell transplantation. Pituitary cell transplantation therapy has not yet been achieved, and is currently under development for future implementation. This study provides statistical data to derive reference figures for transplantation. The progress of our research to date is described below:

We have established a technique to generate pituitary-hypothalamic organoids (PHOs) using feeder cell-free human pluripotent stem cells (hPSCs) for use in human clinical practice and have demonstrated the efficacy of cell transplantation therapy of pituitary organoids in a mouse model of hypopituitary function. Currently, we are conducting cell transplantation experiments in large mammals to verify the efficacy of cell transplantation therapy. To prepare for future pituitary cell transplantation therapy in humans, we aimed to determine the optimal dosage of cells for transplantation. As the transplantable cells we created possess the ability to adjust their secretion in response to the surrounding environment, the amount of basic ACTH secreted by the transplantable cells is not necessarily proportional to their maximum secretory capacity. For instance, even if we transplant twice the number of cells, basal secretion is maintained within a certain range owing to feedback mechanisms. To accurately measure the ability of transplanted cells, it is advisable to include a stimulation test to assess the maximum secretory capacity, thereby avoiding potential overdosing of the cell preparations. The maximum secretory capacity of ACTH in the transplanted cells was evaluated by performing the CRH stimulation test in an experimental mouse model. However, the maximum secretory capacity of ACTH in humans remains unknown. In the present study, we conducted a retrospective analysis of the maximal secretory capacity of ACTH and GH in patients with pituitary adenomas who were determined to have no impairment in ACTH and GH secretory capacity after undergoing CRH and GHRP-2 loading tests. The results indicated that the mean ACTH peak value in the CRH stimulation test was 97.2 pg/ml and the mean GH peak value in the GHRP-2 stimulation test was 25.1 ng/ml. The mean ratio of the ACTH to GH peak values after stimulation was 1:258.2. We propose that these data will serve as realistic indicators for future dose testing in human clinical practice.

Reviewer 3 Report

Comments and Suggestions for Authors

This is an extremely confusing manuscript. The authors aim to define upper limits of ACTH and GH secretion in patients with pituitary adenomas, following stimulation testing with CRH and GHRP-2. However, the majority of the elongated introduction describes work performed on creating pituitary organdies from various sources as a potential therapy to treat hypopituitarism, which is then completely irrelevant to the actual studies performed. At best, this manuscript needs to be significantly re-written to avoid confusion.

Specific concerns:

1) The introduction is far too long, and not remotely relevant to the actual study performed.

2) There is no description of ethical approval for this study.

3) There is no description of the stimulation protocol for this study.

4) The assays are not described, no quality control data are provided, no analyses methods are provided.

Author Response

Reviewer 3

Comments 1. The introduction is far too long, and not remotely relevant to the actual study performed.

Response 1. We appreciate your observation and have subsequently revised the Introduction.

The hypothalamic-pituitary system regulates endocrine functions essential for survival. Hypo-thalamic hormones are secreted into the hypothalamic portal system and directly activate cell surface receptors in the anterior pituitary. Differentiated pituitary cells secrete six hormones: adrenocorticotropic hormone (ACTH, from corticotroph cells), growth hormone (GH, from somatotroph cells), prolactin (PRL, from lactotroph cells), thyroid-stimulating hormone (TSH, from thyrotroph cells), follicle-stimulating hormone and luteinizing hormone (FSH and LH respectively, both from gonadotroph cells). The hypothalamic-pituitary-target axis is crucial for regulating pituitary hor-mone secretion and homeostasis. Corticotropin-releasing hormone (CRH) from the hypothalamus induces ACTH secretion from corticotroph cells in the pituitary. ACTH stimulates cortisol secretion from the adrenal glands, forming an example of negative feedback regulation that inhibits the release of CRH and ACTH. Pituitary dysfunction can occur either congenitally or be acquired. The significance of hypopituitarism can-not be overstated, with 45.5 cases per 100,000 population, indicating a substantial bur-den on public health [1]. Hypopituitarism does not resolve spontaneously, and its treatment primarily relies on hormone replacement therapy (HRT) administered orally. Particularly important is the management of adrenal crisis, a potentially life-threatening complication resulting from inadequate adrenal steroid replacement in adrenocorticotropic hormone (ACTH) deficiency. On the other hand, excessive adrenal steroid replacement can lead over time to iatrogenic Cushing's syndrome. Current treatment modality of HRT thus fails to track adequately the finely tuned hormonal fluctuations of the body, potentially leading to recurrent slight deficiencies or excesses [2], which may contribute to a risk of sudden death greater than that of healthy individuals.

Growth hormone (GH) primarily promotes somatic growth during childhood and is involved in various aspects of metabolic regulation during adulthood. Children with insufficient GH secretion experience less linear growth and exhibit shorter stature than their peers. Adults with GH deficiency develop alterations in body composition, metabolic abnormalities such as dyslipidemia, fatigue, reduced stamina, and decreased concentration, all of which negatively impact their quality of life. Increased dyslipidemia, thickening of carotid intima-media thickness, and a higher incidence of non-alcoholic fatty liver disease have been reported, all of which increase the risk of atherosclerosis [3]. GH replacement therapy is administered for these conditions; however, adherence is often suboptimal, particularly among adults, due to the necessity for self-administration via injection [4]. Our final objective is to develop pituitary hormone-producing cells that respond to environmental cues like those that the hu-man body supplies and to offer therapy more effective than current HRT. 

We have been continuously working to refine differentiation-induced pituitary cells in three-dimensional culture into a technique suitable for clinical use in humans. In 2023, we established a feeder-free pituitary-hypothalamic organoids (PHOs) generation technique using hPSCs with clinical application in mind, achieving nearly 100% success in producing pituitary-hypothalamic tissues, including ACTH-producing cells [5]. PHOs transplanted into a mouse model of hypopituitarism successfully engrafted, im-proved blood ACTH levels, and responded to stimulatory and inhibitory signals [6]. Similar results were observed in purified pituitary cell preparations derived from PHOs. Although hPSC-derived pituitary organoids hold promise for regenerative medicine, several challenges to their implementation remain, one of which is deter-mining the appropriate cell dosage for trans-plantation into humans. Because trans-planted cells can adjust their secretion in response to the environment, baseline secretion levels may not reflect the capacity of the cells. It is desirable to perform stimulation tests to evaluate the secretory capacity of transplanted cells. We conducted CRH stimulation testing in mouse experimental models to assess the maximum secretory capacity of ACTH, but the maximum secretory capacity of ACTH in humans remains unknown. In this study, we retrospectively investigated the maximum secretory capacity of ACTH and GH in patients with pituitary adenomas who exhibited normal ACTH and GH secretion in response to CRH and growth hormone-releasing peptide-2 (GHRP-2) stimulation tests.

Comments 2. There is no description of ethical approval for this study.

Response 2. Ethical approval for this study is described in the Institutional Review Board Statement.

The study was conducted in accordance with the Declaration of Helsinki, and approved by the Ethics Review Committee of our institution (2021-0443 and February 22, 2022).

Comments 3. There is no description of the stimulation protocol for this study.

Response 3. We appreciate your observation and have subsequently revised the Methods.

The CRH stimulation test was conducted in the morning (no later than 10:00 AM) in a fasting state. Following approximately 30 minutes of rest in a supine position, the test was administered while maintaining the same posture. Human CRH (Human CRH for Intravenous Injection "Tanabe", Nipro ES Pharma Co., Ltd., Osaka, Japan) 100 μg was reconstituted in 1 mL of diluent and administered intravenously over approximately 30 seconds. Plasma ACTH levels were measured prior to stimulation and at 30, 60, 90, and 120 min post-stimulation. Serum cortisol levels were measured prior to stimulation and at 30, 60, 90, and 120 min post-stimulation. Similarly, the GHRP stimulation test was conducted in the morning (no later than 10:00 AM) in a fasting state. Following approximately 30 minutes of rest in a supine position, the test was administered while maintaining the same posture. GHRP-2 (GHRP Injection Kaken 100®, Kaken Pharmaceutical Co., Ltd., Tokyo, Japan) 100 μg was reconstituted in 10 mL of physiological saline provided with the preparation and administered intravenously over approximately 30 seconds. Serum GH levels were measured prior to stimulation and at 15, 30, 45, and 60 min post-stimulation.

Comments 4. The assays are not described, no quality control data are provided, no analyses methods are provided.

Response 4. We appreciate your observation and have subsequently revised the Methods.

Plasma ACTH/serum cortisol levels and serum GH/blood IGF-1 levels obtained from the aforementioned methods were determined using electrochemiluminescence immunoassay (ECLIA) kits clinically utilized in Japan (SRL Inc., Tokyo, Japan).